# Effect of Family-Based REDUCE Intervention Program on Children Eating Behavior and Dietary Intake: Randomized Controlled Field Trial

**DOI:** 10.3390/nu12103065

**Published:** 2020-10-08

**Authors:** Norliza Ahmad, Zalilah Mohd Shariff, Firdaus Mukhtar, Munn-Sann Lye

**Affiliations:** 1Department of Community Health, Faculty of Medicine and Health Science, Universiti Putra Malaysia, Serdang 43400, Malaysia; lyems9@yahoo.com; 2Department of Nutrition and Dietetics, Faculty of Medicine and Health Science, Universiti Putra Malaysia, Serdang 43400, Malaysia; zalilahms@upm.edu.my; 3Department of Psychiatry, Faculty of Medicine and Health Science, Universiti Putra Malaysia, Serdang 43400, Malaysia; drfirdaus@upm.edu.my

**Keywords:** child, parents, vegetables, snacks, fruit, overweight, obesity, social media, enjoyment of food, food responsiveness, satiety responsiveness, food record, feeding behavior, diet, sugar-sweetened beverages

## Abstract

The objective of this study was to evaluate the effect of a family-based intervention program (REDUCE) on children’s eating behaviors and dietary intake. A two-arm randomized controlled field trial was conducted among parents and children of 7 to 10 years old who were either overweight or obese. The intervention was conducted via face-to-face sessions and social media. The child eating behaviors were assessed using the child eating behaviors questionnaire (CEBQ), while their dietary consumption of vegetables and unhealthy snacks was assessed using a parental report of three days unweighted food. The generalized linear mixed modelling adjusted for covariates was used to estimate the intervention effects with alpha of 0.05. A total of 122 parents (91% response rate) completed this study. At the six-month post-training, there were statistically significant mean differences in the enjoyment of food (F(6481) = 4.653, *p* < 0.001), fruit and vegetable intake (F(6480) = 4.165, *p* < 0.001) and unhealthy snack intake (F(6480) = 5.062, *p* < 0.001) between the intervention and wait-list groups; however, it was not clinically meaningful. This study added to the body of knowledge of family-based intervention that utilized social media and assessed the effect in children’s eating behavior using the CEBQ and children’s dietary intake.

## 1. Introduction

Childhood obesity is increasing worldwide [1]. In Malaysia, there was a threefold increase in prevalence within a four-year period between 2011 and 2015, 3.9% and 11.9%, respectively [2,3]. Contributory factors to this increasing prevalence in Malaysia include increased access and intake of energy-dense food, sugar-sweetened beverages, increasing sedentary lifestyles and less physical activity. Furthermore, parents often neglected their roles in monitoring their children’s meal intake, especially among families where both parents are working [4]. Parents’ roles are important in promoting healthy diets and availability of healthy food and beverages at home. Moreover, parents can influence their children’s behaviors [5]. A systematic review showed that parents who practiced restrictive guidance or setting rules were negatively associated with unhealthy food intake among children aged seven and older [6]. Parental control of availability of healthy food and unhealthy food at home showed to be the strongest associations with both healthy and unhealthy food intake among children [6]. A meta-analysis which involved 42 weight-related health interventions, showed that interventions that required parents’ participation were more effective in reducing the body mass index (BMI) of the child [7]. Studies including a review have shown an association between children’s eating behaviors and obesity [8,9,10,11]. The child eating behaviors that were defined in these studies include eight constructs which were measured using the children’s eating behavior questionnaire (CEBQ) [12]. The enjoyment of food and food responsiveness were found to be positively associated with weight gain [9,11]. Meanwhile, satiety responsiveness was negatively associated with weight gain [9,10]. To date, assessing the change in children’s eating behaviors following intervention is still scarce. To our knowledge, no intervention study has been conducted to assess the changes in children’s eating behavior except for one study that was conducted among 4 to 7-month-old children [13]. The findings showed no significant group differences in satiety responsiveness (*p* = 0.68) and food responsiveness (*p* = 0.98).

Furthermore, the effect of intervention on consumption of certain energy dense foods and beverages which could possibly explain the reduction in a child’s adiposity, need to be evaluated. A systematic review showed moderately strong evidence that there is a positive association between dietary energy density and increased adiposity in children and adolescents [14]. Energy-dense foods are less satiating than energy-dilute foods. Therefore, energy-dense foods tend to be overeaten. Energy density influences the palatability of food, which then influences consumption [15]. The intake of low-energy dense foods (such as fruits and vegetables) helps to reduce the consumption of high-fat and high-sugar foods among families at risk of childhood obesity [16] by controlling hunger [17]. Although only a few studies showed an inverse relationship between fruit and vegetable intake and body weight, it was recommended that fruits and vegetables replace foods high in energy density among children [18]. Another important determinant of weight gain is sugar-sweetened beverages (SSBs) which have received attention in many obesity-prevention strategies. Recent systematic reviews and meta-analyses of cohort and intervention studies have demonstrated the positive association between SSB intake and weight gain [19,20]. Thus, reduction in SSB intake is important in reducing weight.

Currently, in Malaysia, only morbidly obese children who are detected through school health services are referred to a health center for further management [21]. Those who are overweight and obese with no complications are not referred. Therefore, we developed the REDUCE (REorganise Diet, Unnecessary sCreen time and Exercise) family-based intervention program that targeted parents of overweight and obese children recruited at school. The main aim of the REDUCE program was to reduce the children’s adiposity. The effect of the REDUCE intervention on child adiposity was reported earlier [22]. The intervention group had a significant reduction in BMI z-scores, waist circumference percentile and percentage of body fat compared to wait-list group. The aim of this article is to report the effect of the REDUCE intervention on children’s eating behaviors and dietary intake at baseline, immediate post-training, and three- and six-months post-intervention. We hypothesized that compared to the wait-list group, children in the intervention group will have a reduction in enjoyment of food and food responsiveness but increase in satiety responsiveness as well as a reduction in SSBs and unhealthy snack intake and increase in fruit and vegetable intake. Children’s eating behaviors and dietary intake might contribute to the reduction in children’s adiposity.

## 2. Materials and Methods

A detailed elaboration of the methodology of this field trial has been published elsewhere [22]. However, a summary of the methodology is described here.

### 2.1. Participants and Design

All five primary government schools in an urban community in a state in Malaysia were selected for this study. Parent–child participants were recruited from August until October 2014. Inclusion criteria were parents who have at least one child aged 7 to 10 years with a BMI z-score higher than 1SD, Malay ethnicity, computer literate, had access to the internet and were willing to use social media for interaction. Eleven- and 12-year-old primary school students were not recruited in this study as they will be involved in a national examination during the span of this intervention study. Overweight is classified as a BMI z-score higher than 1SD and obesity when the BMI z-score is higher than 2SD. Height and weight of the children were measured at the screening stage in order to calculate their BMI z-score. Parents were excluded if they reported children having co-morbidities, chronic diseases, physical disabilities, learning disabilities for example autism and attention deficit hyperactive disorder, or on medication for chronic illness. Parents were randomized into intervention and wait-list control groups in a ratio of 1:1. Individual randomization was used, as contamination was thought to be unlikely among parents and the intervention was delivered to parents mainly using online communication. Furthermore, parents in the intervention group were informed not to share the educational materials with other parents as it was at the experimental phase.

### 2.2. Intervention

The four-month REDUCE (REorganise Diet, Unnecessary sCreen time and Exercise) intervention program comprised two phases i.e., training phase for four weeks and booster phase for twelve weeks. It was delivered using face-to-face session and social media. The intervention was based on the social cognitive theory (SCT) [23,24] where parents could influence children’s eating and physical activity behaviors. Table 1 shows the content of REDUCE module addressing children’s eating behaviors, dietary intake and physical activity. Children’s eating behaviors such as enjoyment of food, food responsiveness and satiety responsiveness were addressed in “Parenting skills and roles” and “Problematic scenarios and suggested solutions” in each respective unit. The five targets of the REDUCE program were (i) intake of five servings of fruit and vegetables i.e., two servings of fruits and three servings of vegetables; (ii) no SSB intake; (iii) no unhealthy snack intake; (iv) at least 30 min of moderate to vigorous physical activity; and (v) a maximum of two hours of screen time.

Table 2 shows the application of the theory’s behavior change techniques in the REDUCE module. Parents are trained to manage their children’s eating behaviors, dietary intake, physical activity and screen time using goal setting, self-monitoring, behavioral capability, problem solving, self-efficacy, stimulus control and relapse prevention techniques.

The four-week training phase was delivered through half-day face-to-face sessions (session one) for the first week followed by weekly Facebook sessions for two weeks, and via half-day face-to-face sessions (session two) for the final week (Table 1). Session one in face-to-face training was delivered to parents only, as this was the introductory session that parents need to be clear on their role in educating and guiding their children. Parents were free to clarify any misunderstanding. The second session in the face-to-face training was delivered to parents and children. Children were involved in order to obtain their participation in this intervention.

The twelve-week booster phase took place immediately after the training phase and was conducted weekly using the parents’ dedicated WhatsApp group. It provided parents with a gist of the training module and acted as a platform for interaction and problem solving between researchers and parents. The intervention was mainly facilitated by a public health physician with a total contact time with participants of 22 h. The wait-list control group had access to usual care and received the intervention after the completion of the final 6-month data collection.

### 2.3. Measures

The primary outcome was BMI z-score with secondary outcomes of waist circumference percentile and percentage of total body fat as reported earlier [21,22]. Child behaviors that were assessed include children’s eating behaviors and dietary consumption that are presented here. The children’s eating behaviors were assessed using the Children’s Eating Behavior Questionnaire (CEBQ) which was developed and validated by Wardle et al. (2001) [12]. This questionnaire consists of eight constructs to measure food responsiveness, enjoyment of food, emotional overeating, satiety responsiveness and desire to drink, slowness in eating, emotional undereating and food fussiness. It is a parent-administered questionnaire, comprising 35 items, with 5 Likert scale responses: never (1), seldom (2), sometimes (3), often (4) and always (5).

An exploratory factor analysis was conducted for construct validity of this questionnaire among 160 written consented local parents who were not involved in this trial. Most of the items were loaded on the expected factors and were comparable with their original factors; therefore, a confirmatory factor analysis was not conducted. A study by Santos et al. (2011) found that four behaviors are associated with weight gain i.e., food responsiveness, enjoyment of food, satiety responsiveness and emotional overeating [9]. However, the REDUCE program did not cover emotional overeating in its training contents. Thus, the researcher focused on three behaviors in order to examine the effectiveness of the intervention on “food approach” components of children’s eating behavior, i.e., food responsiveness, enjoyment of food and satiety responsiveness. The test-retest reliability for food responsiveness, enjoyment of food and satiety responsiveness were 0.83, 0.87 and 0.85, respectively. Internal consistencies for those behaviors were 0.85, 0.80 and 0.61, respectively.

For dietary consumption, parents were requested to record their children’s food and beverage intake using a three-day unweighted food record: two weekdays and one day at the weekend. It is an open-ended instrument with no limit on the number of items that can be reported. The information on children’s intake of SSBs, fruit and vegetables and unhealthy snacks was retrieved based on the parents’ reports on the amount and frequency per day. This information then was averaged for the three days for each data collection.

Covariates included sociodemographic information (child’s age, child’s gender, parents’ education and family income) and parents’ body mass index that were collected at baseline using a standard questionnaire. Parents’ body mass index was calculated based on self-reported weight and height.

### 2.4. Data Collection

Data were collected using self-administrated questionnaires which were distributed to the parents via their children at the respective schools. A designated teacher of each school helped to collect the questionnaires once completed.

### 2.5. Ethical Considerations

All parents provided written consent to participate in this study including consent on behalf of their children. The respondents’ information and a consent form that clearly explained the study were given to the parents prior to recruitment. This study was approved by the Medical Research Ethics Committee for Human Research, Universiti Putra Malaysia (Reference: UPM/TNCPI/RMC/1.4.18.1 (JKEUPM)/F2). The findings of this study were to be reported collectively and no names would be revealed. The trial was registered at the Australian New Zealand Clinical Trials Registry: ACTRN12617000844347 (7 June 2017 retrospectively registered) and the National Medical Research Register, Ministry of Health Malaysia: NMRR-14-685-21874 (July 2014).

### 2.6. Statistical Analysis

Calculation of sample size was described elsewhere [21]. We aimed to have 56 parents per arm to detect a significant difference in the BMI z-score between intervention and wait-list control groups of 0.24 with a standard deviation of 0.48 based on a previous study [25], and the dropout rate after randomization was assumed to be 15%. The sample size was not calculated based on the intention to detect a significant difference in children’s eating behaviors because to date, there are no known intervention studies that have been conducted to assess such differences among primary school children.

Data were analyzed using IBM SPSS version 22.0 (Serdang, Selangor, Malaysia). Tests for normality were conducted before further analyses. Non-normally distributed data were log transformed before further analysis. Differences between intervention and wait-list control at baseline were tested using an independent t-test (for continuous variables) or chi-square test or Fischer’s exact test (for categorical variables) to examine the homogeneity between groups. All follow-up outcomes were analyzed using the intention-to-treat principle, in which all study subjects were retained in the groups to which they were originally allocated and no subjects were removed from the analyses, regardless of their adherence to study protocol. Differences of means at each time point in children’s eating behaviors and dietary consumption were compared between intervention and wait-list control groups using an independent t-test for continuous variables. The effectiveness of the intervention was evaluated using generalized linear mixed modelling (GLMM) adjusted for baseline covariates. Covariates included were variables that could potentially affect body weight—child’s age, child’s gender, parents’ BMI, parents’ education and family income. The level of significance was set at α = 0.05, and the null hypothesis rejected when *p* ≤ 0.05.

## 3. Results

One hundred and twenty-two parents completed this study with a response rate of 91% among parents. The Consolidated Standards of Reporting Trials (CONSORT) flow diagram of parents’ participation throughout the study period is shown in Figure 1.

Table 3 shows that there was no significant difference between the intervention and wait-list control groups for the categorical and continuous variables (*p* > 0.05) at baseline.

### Effectiveness of the Intervention on Children’s Eating Behaviors

Table 4 shows the GLMM results for children’s eating behaviors. There was a significant group and time interaction for enjoyment of food. Post-hoc comparison analysis using sequential Bonferroni correction was performed to determine the mean differences of the enjoyment of food score across time within the intervention and within wait-list group. The results showed significant mean differences in the intervention group in the following pairwise comparisons: between baseline and 3-month post-training (*p* = 0.012), baseline and 6-month post training (*p* = 0.008), immediate and 3-month post-training (*p* < 0.001), immediate and 6-month post-training (*p* < 0.001). The decrease in enjoyment of food scores in the intervention group were as we hypothesized. On the other hand, in the wait-list group, none of the pairwise contrasts had significant mean differences.

Table 5 shows the GLMM results for dietary intake per day. There were significant group and time interactions for fruit and vegetable intake and unhealthy snack intake. Post-hoc comparison analysis using sequential Bonferroni correction was conducted to examine the mean differences of fruit and vegetable intake across time within the intervention and within wait-list group. The analyses showed statistically significant slight decrease in fruit and vegetable intake in the intervention group from 3-month post-training to 6-month post-training (*p* = 0.036). On the contrary, in the wait-list group, there were statistically significant mean differences in the following comparisons: decrease in fruit and vegetable intake from baseline to immediate post-training (*p* = 0.033), increase in fruit and vegetable intake from immediate post-training to 3-month post-training (*p* = 0.003) and decrease in fruit and vegetable intake from 3-month post-training to 6-month post-training (*p* = 0.019). The decrease in the mean in the fruit and vegetable intake was contrary to our hypothesis.

The results of post-hoc comparison analyses on unhealthy snacks showed significant mean differences in the intervention group in the following comparisons: baseline and immediate post-training (*p* = 0.004), immediate post-training and 3-month post-training (*p* < 0.001) and immediate post-training and 6-month post-training (*p* < 0.001). The reduction in the mean was only at immediate post-training. There were no significant mean differences for pairwise comparisons in the wait-list group.

## 4. Discussion

Studies have shown an association between children’s eating behaviors and dietary consumption with obesity [8,9,10,11]. However, the effect of parental intervention in altering children’s eating behaviors and dietary consumption on child obesity is still lacking. This is among the few randomized controlled field trials (RCT) that evaluated the changes in children’s eating behaviors and dietary intake following family-based intervention. In summary, there were small changes in enjoyment of food, consumption of fruits and vegetables and unhealthy snacks.

In the present study, children in the intervention group were observed to have a significant decreased in enjoyment of food compared to the wait-list control group. Although there was a statistically significant difference, the effect was too small to have any clinical impact. This study did not find any significant changes in satiety responsiveness and food responsiveness. New strategies in parental intervention need to be explored in order to observe clinically significant changes in children’s eating behaviors. To date, there has been no RCT of children in the same age group using the CEBQ, so there is limited scope for comparison. An RCT used the CEBQ to evaluate a 12-week protective feeding intervention delivered to children when they were aged 4 to 7 months and again at 13 to 16 months [13]; the control group received standard care. The RCT involved 698 mother–infant dyads who were assessed when their children were aged 2, 3.7 and 5 years. There was no significant group and time interaction between the intervention and control group in their study as well.

In the present study, no statistically significant interaction was found despite the sustained reduction in SSB intake in the intervention group at the six-month follow up. There is a possibility that this may be due to a type 2 error. It could possibly have been significant if the sample size was larger and had sufficient power. Most internet-based family interventions did not measure SSB consumption [25,26,27,28,29]. The exception was a study by Knowlden (2013) that reported no difference in SSB consumption between the intervention and control groups [30]. Some conventional family-centered intervention studies have measured SSB consumption as an outcome variable [31,32,33,34,35,36,37,38,39]. One conventionally delivered family intervention which measured SSB intake [40] found no group difference in consumption (intervention M = 1.47, SE = 0.06; control M = 1.56, SE = 0.06; *p* = 0.49), although another study [41] reported a small group difference in change in consumption of carbonated soft drinks at the six-month follow-up (intervention M = −0.3 servings/day; control M = −0.1 servings/day; *p* = 0.02). The intervention consisted of regular household visits by health counsellors to promote a change in lifestyle to 57 households comprising a total of 174 individuals aged 5 to 70 years, while the control group received standard care. Another RCT found that a six-month parental support program, ”Healthy School Start”, reduced consumption of unhealthy drinks, but the effect was not sustained at a five-month post-intervention follow-up (*p* = 0.83) [42].

At baseline, the mean fruit and vegetable intake was in the range of one intake per day, and there had not been much change at the end of the study period. This figure is still well below the target of five servings per day. The Malaysian Dietary Guidelines for Children and Adolescents (2013) has suggested that children and adolescents aged seven to 18 years should consume at least two servings of fruit and three servings of vegetables per day to improve overall health and reduce the risk of certain non-communicable diseases [43]. Knowlden (2013) found a higher increase in consumption of fruit and vegetables in the intervention group (baseline: M = 3.83, SD = 1.78; eight-week follow-up: M = 6.04, SD = 2.8) [30] compared to increase in consumption in the control group (baseline M = 2.95, SD = 1.58; eight-week follow-up M = 3.70, SD = 1.51). However, his study only involved a short follow-up period of eight weeks. Other intervention studies also did not achieve the five servings of fruit and vegetables per day [27,41,42].

A review of quantitative research on determinants of fruit and vegetable intake concluded that taste was the main reason for children’s dislike of vegetables; vegetables were associated with unpleasant and negative tastes such as bitterness, sourness and blandness [44]. This review also concluded that children disliked fruit because of poor appearance e.g., bruising, which was in turn associated with unsatisfactory taste. Fruit and vegetables were not perceived to be convenient snack foods as they were not instantly available and had to be washed, peeled, cut or cooked before consumption, and parents were expected to do this for their children. This meant that although fruit and vegetables may be available in homes, in practice they are not accessible to children. It is also easy for fruit and vegetables to get squashed and soggy in children’s school bags. Unhealthy snacks were perceived as much tastier, more convenient and more accessible. It is important for researchers to find ways of overcoming these barriers in order to increase parents’ and children’s preferences for fruit and vegetables and improve their availability and accessibility in the home. Achieving these goals might increase families’ fruit and vegetable intake to the recommended level.

This study found a significant decrease in intake of unhealthy snacks at immediate post-training in the intervention group, followed by an increase in intake at the 3- and 6-month post-intervention. This showed that the intervention did not produce a sustained effect on the unhealthy snacks’ intake. Similarly, a school-based intervention study involving parents resulted in a lower intake of unhealthy foods (snacks, ice cream, cookies and sweets) (*p* = 0.01) in the intervention group compared to the control group immediately after the intervention, but the effect was not maintained at the five-month post-intervention assessment [42]. Despite the length of the six-month intervention, which involved parents (given health information and two 45-min motivational interviews) and teachers (ten sessions of 30-min classroom activities with teacher), it had no sustained effect on children’s intake of unhealthy snacks. More effort should be made to reduce children’s unhealthy snack consumption, as this is among the important factors contributing to obesity. Many factors influence intake of unhealthy snacks, and media advertising is thought to play an important role [45] alongside availability of unhealthy snacks outside schools at cheap prices that make them easily accessible and readily consumed by children, especially among the lower socio-economic group [46]. The socio-cultural environment in Malaysian schools also promotes consumption of unhealthy snacks, as schools use sweet foods and drinks as rewards at large events [47]. Previous Internet-based family interventions and conventionally-delivered family interventions did not measure children’s intake of unhealthy snacks.

The limitations of the present study include self-reporting in questionnaires involving recording of food that is not weighted. This may result in underreporting or over reporting especially with regard to children’s intake of food and beverages. There is also a possibility that the children might have filled in the questionnaire themselves. The sample size may not have had sufficient power to detect a clinically meaningful effect of intervention on SSB intake as significant.

## 5. Conclusions

This four-month REDUCE intervention program aimed to improve children’s eating behavior and dietary intake in order to reduce children’s BMI z-score. The findings of this RCT showed significant small mean differences in the enjoyment of food, fruit and vegetable intake and unhealthy snack intake between the intervention and wait-list groups. However, these differences were not clinically meaningful. This was the first family-based intervention study that combined face-to-face sessions and social media and the first RCT to assess the changes in children’s eating behaviors using the CEBQ among primary school children. The findings of this study added to the body of knowledge of family-based intervention studies on childhood obesity.

## Figures and Tables

**Figure 1 nutrients-12-03065-f001:**
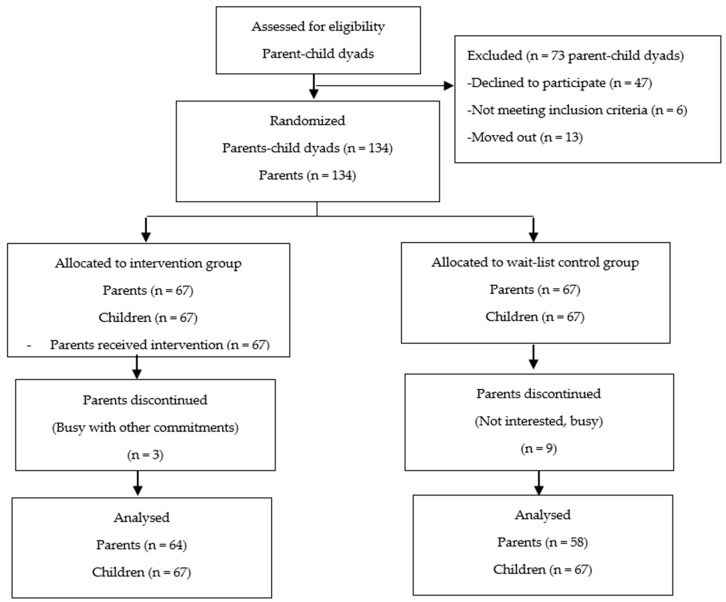
Study flow chart and Consolidated Standards of Reporting Trials (CONSORT) diagram of parent–child participation of REDUCE (REorganise Diet, Unnecessary sCreen time and Exercise) randomized controlled field trial.

**Table 1 nutrients-12-03065-t001:** Contents of REDUCE (REorganise Diet, Unnecessary sCreen time and Exercise) module which were delivered over a four-week period for the intervention group.

Unit	Topic and Contents in REDUCE Program	Unit	Topic and Contents in REDUCE Program
1	**Introduction and parenting skills *** • Introduction of program • Obesity and healthy lifestyles • Parenting skills and parental role • How to fill in food record • Information on REDUCE program	2	**Sugar-sweetened beverages (SSB) *** • Target for SSB • What is SSB? • Why are SSBs bad for health? • Tips to reduce SSB intake • Parenting skills and roles • Problematic scenarios and suggested solutions • SSB Diary • Feedback and discussion during face-to-face training
3	**Fruit and vegetable (FV) **** • Target for FV • Why FV are good for health? • Example of servings for fruit and vegetables • How to cook vegetables • Tips to increase FV intake • Parenting skills and roles • Problematic scenarios and suggested solutions • SSB and FV Diary • Feedback and discussion on social media	4	**Unhealthy snacks/junk food **** • Target for snacks • What are snacks? • Why are snacks bad for health? • Types of fat • Reading food labels • Tips to reduce snack intake • Parenting skills and roles • Problematic scenarios and suggested solutions • Food record • Feedback and discussion on social media
5	**Physical activity **** • Target for physical activity • What is physical activity? • Why is physical activity good for health? • Examples of moderate and vigorous activities • Examples of activity to strengthen muscles and bones • Tips to increase physical activity • Parenting skills and roles • Problematic scenarios and suggested solutions • Physical activity record • Feedback and discussion on social media	6	**Screen time **** • Target for screen time • What is screen time? • Why prolong screen time bad for health? • Tips to reduce screen time • Parenting skills and roles • Scenario of problem and suggested solutions • Physical activity record including screen time • Feedback and discussion on social media
7	**Risky situations and review of performance *** • What are “risky situations” • Examples of risky situation • Why are these risky situations bad for health? • How to deal with “risky situations” • Parenting skills and roles • Problematic scenarios and suggested solutions • Feedback and discussion during face-to-face training	8	**Further role and action *** • Obesity issues • Exercise Tips *** • Calories needed and examples of serving according to age and sex • Examples of success stories • Summary of REDUCE program • Parental role

Notes: * Face-to-face approach, ** social media approach, *** delivered by a sport medicine specialist.

**Table 2 nutrients-12-03065-t002:** The application of Social Cognitive Theory (SCT)’s behavior change techniques in the REDUCE module.

SCT’s Behavioral Change Techniques	REDUCE Program
Goal setting	Units 2, 3, 4, 5 and 6: These units explain about the five targets * and parents are expected to set goals that are suitable after discussion with their children. Parents are encouraged to choose one goal and make small changes, one at a time.
Self-monitoring	Units 2, 3, 4, 5 and 6: Parents are encouraged to monitor the achievement of the goals * that they set using the booklet which was provided to them.
Problem solving	Units 2, 3, 4, 5, 6 and 7: Parents are informed about problems that may arise in achieving the goals * and examples of how to manage them. Parents are encouraged to write down any problems they have encountered and how they have managed them in the booklet. They are also encouraged to communicate with the researcher about the problems through Facebook or WhatsApp if they are unable to solve them.
Behavioral capability	Unit 1 to Unit 8: Parents are expected to gain the knowledge and skills needed to achieve the goals *. Parents are also equipped with authoritative parenting skills to handle their reluctant children. Helping parents to handle the problems they face may improve their self-efficacy. Parents are also provided with feedback on their children’s anthropometric progress.
Stimulus control	Units 2, 4 and 7: Parents are taught to avoid or control the environment, either outside or inside the house that might induce intake of unhealthy foods.
Relapse prevention	Unit 8: Parents are taught about high-risk situations where it is difficult for them to achieve the goals * and how to handle them.

Notes: * Five targets or goals include (i) five servings of fruit and vegetables, (ii) no sugar-sweetened beverages (SSB), (iii) no unhealthy snacks, (iv) a minimum of 30 min of moderate to vigorous physical activity and (v) a maximum of 120 min of screen time.

**Table 3 nutrients-12-03065-t003:** Baseline characteristics of 134 parent-dyads allocated to the intervention or wait-list control group.

Characteristics	Mean (SD) or *N* (%)		
	Intervention	Wait-List Control	*p*-Value
**Parent**			
Age (years)	39.8 (3.6)	41.3 (5.7)	0.079
Gender—female ^a^	39 (58.2)	37 (55.2)	0.862
BMI (kg/m^2^) ^b^	27.4 (4.41)	27.8 (4.27)	0.622
Mother’s education ^a^			0.665
Secondary and below	24 (35.8)	27 (40.3)	
Tertiary	43 (64.2)	40 (59.7)	
Father’s education ^a^			0.419
Secondary and below	20 (29.9)	28 (41.8)	
Tertiary	47 (70.1)	39 (58.2)	
Monthly family income ^a^			0.921
Less than RM5000	21 (31.3)	22 (32.8)	
RM5000 to RM10000	29 (43.3)	30 (44.8)	
More than RM10000	17 (25.4)	15 (22.4)	
**Child**			
Age (years)	9.6 (1.2)	9.6 (1.2)	0.826
Gender—female ^a^	40 (59.7)	38 (56.7)	0.861
BMI z-score	2.0 (0.4)	2.1 (0.4)	0.381
BMI z-score category ^a,b^			0.861
Overweight (%)	28 (41.8)	27 (40.3)	
Obese (%)	39 (58.2)	40 (59.7)	

Data are the mean (standard deviation) or *n* (%). ^a^ Categorical variable. ^b^ World Health Organization standards.

**Table 4 nutrients-12-03065-t004:** Effect of REDUCE intervention program on children’s eating behaviors.

Children’s Eating Behaviors	Parameter	F statistics	Df1	Df2	*p*-Value ^a^
Enjoyment of food	Group	1.167	1	481	0.280
(*n* = 119)	Group × Time	4.653	6	481	<0.001 *
Satiety responsiveness	Group	7.142	1	481	0.008 *
(*n* = 120)	Group × Time	1.430	6	481	0.201
Food responsiveness	Group	0.525	1	475	0.469
(*n* = 118)	Group × Time	2.054	6	475	0.057

^a^ Using generalized linear mixed model adjusted for child’s age, child’s gender, parents’ body mass index, mother’s education, father’s education, family income and score of children’s eating behaviors at baseline. * Significant at *p* ≤ 0.05. Df: degree of freedom.

**Table 5 nutrients-12-03065-t005:** Effect of REDUCE intervention program on dietary intake per day.

Dietary Intake	Parameter	F	Df1	Df2	*p*-Value ^a^
SSB intake (*n* = 122)	Group	9.647	1	480	0.002 *
	Group × Time	1.373	6	480	0.224
Fruit and vegetable intake (*n* = 122)	Group	1.493	1	480	0.222
	Group × Time	4.165	6	480	<0.001 *
Unhealthy snack intake (*n* = 122)	Group	0.166	1	480	0.684
	Group × Time	5.062	6	480	<0.001 *

^a^ Using generalized linear mixed model adjusted for child’s age, child’s gender, parents’ body mass index, mother’s education, father’s education, family income and dietary intake at baseline. * Significant at *p* ≤ 0.05.

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
