# Peer review of "Effect of Family-Based REDUCE Intervention Program on Children Eating Behavior and Dietary Intake: Randomized Controlled Field Trial"

_nutrients, 2020, doi:10.3390/nu12103065_

Round 1

Reviewer 1 Report

This is an accompanying paper to the main  RCT and describes changes in behaviour. As the effects are rather small I think it might be possible to shorten this paper but it is very important that papers with negligible outcomes are always published to avoid publication bias. 

Introduction 

I think that lines 49 to 54 gives too much detail on a previous study please summarise this.

You have an error I think you discuss  EoF - enjoyment of food but at line 45 we have EoE. On page 5 line 136 we have EF?

Methods

line 98 everybody knows that RCTs are meant to distribute confounders equally so the comments are unnecessary. What I do question is the prospect of contamination between arms as you recruited from schools where different families  mix. Why not a cluster design?

Table 1 make it clearer in the heading the time periods for the modules over the intervention.

Table 2 You report five targets in the Table but they should be listed so the table stands alone.

155-164 Too much time was spent on justifying the unweighed food record - this is methods can you make it shorter. The data on relative validity gives the false impression  it was a part of this study.

Results

Can you position the flow chart all on one page.

Section 3.1 disappointing that you have different numbers at the various time points. How does this influence the outcomes which appear to be very minimal. The information in this section is not very helpful in explaining the results.

Table 5 is a good summary of findings. I do not think you need to show the figures as well which give exactly the same information. Exactly the same for Table 7 and Figure 3.

The discussion section is satisfactory.

Author Response

Response to Reviewer 1 Comments

This is an accompanying paper to the main RCT and describes changes in behaviour. As the effects are rather small, I think it might be possible to shorten this paper but it is very important that papers with negligible outcomes are always published to avoid publication bias. 

Introduction 

Point 1: I think that lines 49 to 54 gives too much detail on a previous study please summarise this.

Response 1: Thank you for the comments. We have summarized these. See Page 2, line, 55-59. “To our knowledge, no intervention study was conducted to assess the changes in children’s eating behaviour except for one study that was conducted among 4 to 7 months children [13]. The findings showed no significant group differences in satiety responsiveness (p = 0.68) and food responsiveness (p = 0.98)”.

Point 2: You have an error I think you discuss  EoF - enjoyment of food but at line 45 we have EoE. On page 5 line 136 we have EF?

Response 2: Thank you for the comments. We have removed all abbreviations related to children’s eating behaviour questionnaire ie FR, EF, EOE, SR, DD, SE, EUE and FF to avoid confusion.

Methods

Point 3: line 98 everybody knows that RCTs are meant to distribute confounders equally so the comments are unnecessary.

Response 3: Thank you for the comment. We have deleted the sentence. See Page 3, line 124-126.

Point 4: What I do question is the prospect of contamination between arms as you recruited from schools where different families mix. Why not a cluster design?

Response 4: Thank you for the comments. The reason for using individual design RCT was explained in Page 3, line 127-131 “Individual randomization was used as contamination was thought to be unlikely among parents and the intervention was delivered to parents mainly using online communication. Furthermore, parents in the intervention group were informed not to share the educational materials with other parents as it was at experimental phase”. 

Point 5: Table 1 make it clearer in the heading the time periods for the modules over the intervention.

Response 5: Thank you for the comment. The heading has been changed to “Contents of REDUCE module which were delivered over a four-week period for the intervention group”. See Page 4, line 145.

Point 6: Table 2 You report five targets in the Table but they should be listed so the table stands alone.

Response 6: Thank you for the comment. We have listed the five targets. We put foot note under Table 2 that listed the five targets. See Page 5, Line 153-155.

Point 7: 155-164 Too much time was spent on justifying the unweighed food record - this is methods can you make it shorter. The data on relative validity gives the false impression  it was a part of this study.

Response 7: Thank you for the comments. Justification on the unweighted food record has been removed.

Results

Point 8: Can you position the flow chart all on one page.

Response 8: Thank you for the comment. The flow chart has been placed in one page. See page 8.

Point 9: Section 3.1 disappointing that you have different numbers at the various time points. How does this influence the outcomes which appear to be very minimal. The information in this section is not very helpful in explaining the results.

Response 9: Thank you for your comments. Yes, the results do not show very large differences across time, indicating that, within groups, the fluctuation in scores for the different food/beverages intake is minimal. We have removed the Table 4 and 6 and its respective narration.

Point 10: Table 5 is a good summary of findings. I do not think you need to show the figures as well which give exactly the same information. Exactly the same for Table 7 and Figure 3.

Response 10: Thank you for your comments. Figures 2 and 3 have been removed from the text.

Reviewer 2 Report

Major remarks

Thank you for providing the manuscript. Given the severe impact of childhood and the increasing trend, we still need to learn more about how to prevent and treat obesity in children. Please find below my remarks and suggestion regarding your manuscript:

Introduction, line 35-40: the rationale why focusing on parents for obesity treatment in children could be a bit more elaborated. Which parenting styles/practices work better among obese/overweight children, which aspects of the home food environment have the most impact to create a change in obese/overweight children? Also, relevant seems which determinants are relevant to change parents’ parenting behaviours as that is what the intervention wants to target as well (I assume?).

Line 41-48: The description of the CEBQ does not really belong in an introduction section. I would just focus on relevant constructs when describing earlier studies but including details on measures is too detailed for an introduction. What I find a bit confusing is that you will focus on the age group 7-10-year-old children but in the paragraph on the types of eating behaviours intervention studies are reported that look into effects among infants and toddlers. I would suggest to check the literature on articles regarding the age group that is the focus of this study. In addition, it would be good to add some literature regarding the association between types of eating behaviours and relevant energy-balance related behaviours / dietary behaviours, again in the target age group.

In general, I think the structure of this introduction could be improved. It might be easier to follow if first childhood obesity in Malaysia/Asia/Middle income countries can be described. Followed by causes of obesity regarding dietary patterns, to then make the link to the types of eating behaviours that play a role for both these dietary behaviours as well as for adiposity, and afterwards link it to the environmental impact (i.e. parental practices). I would also expect more elaboration on successful interventions and what your study adds as it is not clear to me what the added value of this study is compared to the current state-of-the-art. Did earlier studies not focus on changing type of eating behaviours via their intervention? Were they not conducted in (upper) middle income countries?

Line 73-85: It would be good to know the effect on child adiposity and not merely referring to [21]. It could then also be relevant to investigate how this effect emerges… Is it the impact on the dietary behaviours that explain the effect on adiposity? And is this change in dietary behaviour mediated by the type of eating behaviours?

Method:

Line 92: What is the rationale behind the target group? Why 7-10-year-olds?

Participants paragraph: Can you give more insight into a sample calculation, how many parent-child dyads were necessary to evaluate the effects. Did you consider clustering of parents from the same school or neighbourhood? A total of 122 parents so about 60 in every condition seems really limited to assess significant and meaningful effects on dietary/eating behaviours? It would help to have a bit more details about where and how questionnaires were completed.

Intervention paragraph: More information on the targeted determinants in parents and which behavioural change techniques are applied to change these determinants is relevant. In table 2 some of the behavioural change strategies are mentioned but what determinants are they targeting. Also, self-efficacy is included but this is not a BCT but a determinant. You also refer to the social cognitive theory but what determinants in this theory are targeted. More elaboration on this is necessary.

Line 152: why was an unweighted food record chosen instead of a weighted one? Would this be so much more work to fill in?

Line 312-314, add references to “studies have shown…”

Minor remarks

  • EOF is not defined in the introduction.
  • It would improve readability if there is a more limited use of abbreviations.

Author Response

Response to Reviewer 2 Comments

Major remarks

Thank you for providing the manuscript. Given the severe impact of childhood and the increasing trend, we still need to learn more about how to prevent and treat obesity in children. Please find below my remarks and suggestion regarding your manuscript:

Point 1: Introduction, line 35-40: the rationale why focusing on parents for obesity treatment in children could be a bit more elaborated. Which parenting styles/practices work better among obese/overweight children, which aspects of the home food environment have the most impact to create a change in obese/overweight children? Also, relevant seems which determinants are relevant to change parents’ parenting behaviours as that is what the intervention wants to target as well (I assume?).

Response 1: Thank you for the comments. We have added information on parenting practices and aspects in home food environment. See Page 1, line 37-44. “Furthermore, parents often neglected their roles in monitoring their children’s meal intake especially among families where both parents are working [4]. Parents’ roles are important in promoting healthy diets and availability of healthy food and beverages at home. Moreover, parents can influence their children’s behaviours (5). A systematic review showed that parents with restrictive guidance or setting rules were negatively associated with unhealthy food intake among children aged seven and older [6]. Furthermore, parental control of availability of healthy food and unhealthy food at home showed to be the strongest associations with both healthy and unhealthy food intake among children [6]”.

Point 2: Line 41-48: The description of the CEBQ does not really belong in an introduction section. I would just focus on relevant constructs when describing earlier studies but including details on measures is too detailed for an introduction. What I find a bit confusing is that you will focus on the age group 7-10-year-old children but in the paragraph on the types of eating behaviours intervention studies are reported that look into effects among infants and toddlers. I would suggest to check the literature on articles regarding the age group that is the focus of this study. In addition, it would be good to add some literature regarding the association between types of eating behaviours and relevant energy-balance related behaviours / dietary behaviours, again in the target age group.

Response 2: Thank you for the comments. We have removed the description of CEBQ in the introduction section. Since to our knowledge, no intervention study was conducted among 7-10-year-old children, we have rephrased the sentence. See Page 2, line 51-59. “The child eating behaviours that were defined in these studies include eight constructs which were measured using children’s eating behaviour questionnaire (CEBQ) [12]. The enjoyment of food and food responsiveness were found to be positively associated with weight gain [9,11]. Meanwhile, satiety responsiveness was negatively associated with weight gain [9,10]. To date, assessing the change in children’s eating behaviours following intervention is still scarce. To our knowledge, no intervention study has been conducted to assess the changes in children’s eating behaviour except for one study that was conducted among 4 to 7 months old children [13]. The findings showed no significant group differences in satiety responsiveness (p = 0.68) and food responsiveness (p = 0.98)”.

Point 3: In general, I think the structure of this introduction could be improved. It might be easier to follow if first childhood obesity in Malaysia/Asia/Middle income countries can be described. Followed by causes of obesity regarding dietary patterns, to then make the link to the types of eating behaviours that play a role for both these dietary behaviours as well as for adiposity, and afterwards link it to the environmental impact (i.e. parental practices). I would also expect more elaboration on successful interventions and what your study adds as it is not clear to me what the added value of this study is compared to the current state-of-the-art. Did earlier studies not focus on changing type of eating behaviours via their intervention? Were they not conducted in (upper) middle income countries?

Response 3: Thank you for the comments. We have amended the introduction section. See Page 1, line34-43, Page 2, line 44-59 and Page 2, line 89-109.

Point 4: Line 73-85: It would be good to know the effect on child adiposity and not merely referring to [21]. It could then also be relevant to investigate how this effect emerges… Is it the impact on the dietary behaviours that explain the effect on adiposity? And is this change in dietary behaviour mediated by the type of eating behaviours?

Response 4: Thank you for the comments. We have made changes in the manuscript. See page 3, line 101-102.  “The intervention group had significant reduction in BMI z-scores, waist circumference percentile and percentage of body fat compared to wait-list group” and line 108-109. “Children’s eating behaviours and dietary intake might contribute to the reduction in children’s adiposity”

Method:

Point 5: Line 92: What is the rationale behind the target group? Why 7-10-year-olds?

Response 5: Thank you for the comments. The recruitment took place towards the end of the year. In Malaysia, the 12 year old children will take a national examination. Therefore, the Ministry of Education has set regulation that this age group will not participate in any study. Hence, when the intervention study took place in the subsequent year, the maximum age will be 11-year-old children. We have included an explanation in the manuscript. See page 3, line 118-119. “Eleven- and 12-year-old primary school students were not recruited in the study as they will be involved in a national examination during the span of this intervention study”.

Point 6: Participants paragraph: Can you give more insight into a sample calculation, how many parent-child dyads were necessary to evaluate the effects. Did you consider clustering of parents from the same school or neighbourhood? A total of 122 parents so about 60 in every condition seems really limited to assess significant and meaningful effects on dietary/eating behaviours?

Response 6: Thank you for the comments. The sample size calculation was based on detecting BMI z-score differences between intervention and wait-list groups using the standard formula for trials using individual randomisation:

  n = 2s2[zα + zβ]2/ (μ1 – μ2)2.

A difference of 0.24 in BMI z-scores with a standard deviation of 0.48 was expected to be relevant, based on a previous study (28). Thus, n = 2(0.48)2 [1.64 + 0.84]2 /(0.24)2 = 49 per group, at 95% level of significance and 80% power. The dropout rate was assumed at about 15% after randomisation; a minimum sample size of 56 parents per group is required to detect this difference with a two-sample t-test. A sentence has been added in the manuscript. See Page 7, line 224-226.  “The sample size was not calculated based on to detect a significant difference in children’s eating behaviours as no previous intervention studies have been conducted in assessing such differences in primary school children”.

Point 7: It would help to have a bit more details about where and how questionnaires were completed.

Response 7: Thank you for the comments. We have added in the manuscript. See Page 6, line 180-181. “The questionnaires were distributed to the parents via their children at respective school. A designated teacher of each school helped to collect the questionnaires once completed”.

Point 8: Intervention paragraph: More information on the targeted determinants in parents and which behavioural change techniques are applied to change these determinants is relevant. In table 2 some of the behavioural change strategies are mentioned but what determinants are they targeting. Also, self-efficacy is included but this is not a BCT but a determinant. You also refer to the social cognitive theory but what determinants in this theory are targeted. More elaboration on this is necessary.

Response 8: Thank you for the comments. We have added targeted determinants/goals. See Page 3, line 140-144 “The five targets of REDUCE programme were (i) intake of five servings of fruit and vegetables i.e. two servings of fruits and three servings of vegetables, (ii) no SSBs intake, (iii) no unhealthy snacks intake, (iv) at least 30 minutes of moderate to vigorous physical activity and (v) maximum of two hours of screen time” and also added the five targeted determinants as foot notes in Table 2. See Page 5. We have included behavioural capability. See Table 2.

Point 9: Line 152: why was an unweighted food record chosen instead of a weighted one? Would this be so much more work to fill in?

Response 9: Thank you for the comments. Yes, the unweighted food record was chosen instead of a weighted one as it is time consuming for the parents and labour intensive. These were explained in the original manuscript. However, based on the comments by Reviewer 1, these explanations have been deleted.

Point 10: Line 312-314, add references to “studies have shown…”

Response 10: Thank you for the comments. We have added the references [8-11]. See Page 13, line 381.

Minor remarks

Point 11: EOF is not defined in the introduction.

Response 10: Thank you for the comments. Abbreviation has been removed.

Point 12: It would improve readability if there is a more limited use of abbreviations.

Response 11: Thank you for the comments. Abbreviations were kept to the minimal.

Round 2

Reviewer 1 Report

Thank you very much for clarifying these points.

The only outstanding problem is at line 266 

'The sample size was not calculated based on to detect a
significant difference in children’s eating behaviours as no previous intervention studies have been conducted in assessing such differences in primary school children.

This is not good English so please correct.'

Also lines 169-181 are no longer formatted correctly.

Author Response

Thank you very much for clarifying these points.

Point 1: The only outstanding problem is at line 266.

'The sample size was not calculated based on to detect a significant difference in children’s eating behaviours as no previous intervention studies have been conducted in assessing such differences in primary school children.

This is not good English so please correct.'

Response 1: Thank you for the comments. We have amended the sentence. “The sample size was not calculated based on the intention to detect a significant difference in the children’s eating behaviours because to date, there are no known intervention studies that have been conducted to assess such differences among primary school children.” See Page 7, line 228 to 231.

Point 2: Also lines 169-181 are no longer formatted correctly.

Response 2: Thank you for the comment. We have formatted the said lines. See Page 6, line 169-180. We have placed the information about the data collection at 2.4 Data collection. See Page 6, line 210-213.
